**On Coupled Unsaturated-Saturated Flow Process Induced by Vertical,**
**Horizontal and Slant Wells in Unconfined Aquifers**
Xiuyu Liang[a*], Hongbin Zhan[b*], You-Kuan Zhang[c], Jin Liu[a]
[a]School of Earth Sciences and Engineering, Nanjing University,
Nanjing, Jiangsu 210093, P.R. China(xyliang@nju.edu.cn)
[b]Department of Geology & Geophysics, Texas A&M University, College Station, TX 77843-
3115, USA. (zhan@geos.tamu.edu)
[c]School of Environment Sciences and Engineering,
South University of Sciences and Technology of China,
Shenzhen, Guangdong 518055, P.R. China
*Co-corresponding authors

19 Revised version submitted to *Hydrology and Earth System Sciences*

20 January, 2017

## Abstract

Conventional models of pumping tests in unconfined aquifers often neglect the unsaturated flow process. This study concerns coupled unsaturated-saturated flow process induced by vertical, horizontal, and slant wells positioned in an unconfined aquifer. A mathematical model is established with special consideration of the coupled unsaturated-saturated flow process and well orientation. Groundwater flow in the saturated zone is described by a three-dimensional governing equation, and a linearized three-dimensional Richards' equation in the unsaturated zone. A solution in Laplace domain is derived by the Laplace-finite Fourier transform and the method of separation of variables, and the semi-analytical solutions are obtained using a numerical inverse Laplace method. The solution is verified by a finite-element numerical model. It is found that the effects of the unsaturated zone on the drawdown of pumping test exist in any angle of inclination of the pumping well, and this impact is more significant for the case of a horizontal well. The effects of unsaturated zone on the drawdown are independent of the length of the horizontal well screen. The vertical well leads to the largest water volume drained from the unsaturated zone ($W$) value during the early time, and the effects of the well orientation on $W$ values become insignificant at the later time. The screen length of the horizontal well does not affect $W$ for the whole pumping period. The proposed solutions are useful for parameter identification of pumping tests with a general well orientation (vertical, horizontal, and slant) in unconfined aquifers affected from above by the unsaturated flow process.

**Keywords:** Horizontal well; Slant well; Coupled unsaturated-saturated flow; Drainage from the unsaturated zone.

# 1. Introduction

In addition to conventional vertical wells, horizontal and slant pumping wells are broadly used in the petroleum industry, environmental and hydrological applications in recent decades. Horizontal and slant pumping wells are commonly installed in shallow aquifers to yield a large amount of groundwater (Bear, 1979) or to remove a large amount of contaminant (Sawyer and Lieuallen-Dulam, 1998). Horizontal and slant wells have some advantages over vertical wells (Yeh and Chang, 2013;Zhan and Zlotnik, 2002), e.g., horizontal and slant wells yield smaller drawdowns than the vertical wells with the same pumping rate per screen length. Horizontal and slant wells have long screen sections which can extract a great volume of water in shallow or low permeability aquifers without generating significant drawdowns.

Hantush and Papadopulos (1962) firstly investigated the problem of fluid flow to a horizontal well in hydrologic sciences. Since then, this problem was not of great concern in the hydrological science community because of the limitation of directional drilling techniques and high drilling costs. With significant advances of the directional drilling technology over the last 20 years, the interest on horizontal and/or slant wells was reignited. Until now flow to horizontal and/or slant wells have been investigated in various aspects, including flow in confined aquifers (Cleveland, 1994;Zhan, 1999;Zhan et al., 2001;Kompani-Zare et al., 2005), unconfined aquifers (Huang et al., 2016;Rushton and Brassington, 2013;Zhan and Zlotnik, 2002;Huang et al., 2011;Mohamed and Rushton, 2006;Kawecki and Al-Subaikhy, 2005), leaky confined aquifers (Zhan and Park, 2003;Sun and Zhan, 2006;Hunt, 2005), and fractured aquifers (Nie et al., 2012;Park and Zhan, 2003;Zhao et al., 2016). The readers can consult Yeh and Chang (2013) for a recent review of well hydraulics on various well types, including horizontal and slant wells.

As demonstrated in previous studies, horizontal and slant wells had significant advantages
over vertical wells in unconfined aquifers, thus they were largely used in unconfined aquifers for
pumping or drainage purposes. However, none of above-mentioned studies considered the effects
of unsaturated processes on groundwater flow to horizontal and slant wells in unconfined
aquifers. For the case of flow to vertical wells in saturated zones, the effects of above unsaturated
processes were investigated by several researchers (Kroszynski and Dagan, 1975;Mathias and
Butler, 2006;Tartakovsky and Neuman, 2007;Mishra and Neuman, 2010, 2011). For example,
Tartakovsky and Neuman (2007) considered axisymmetric unsaturated-saturated flow for a
pumping test in an unconfined aquifer and employed one parameter that characterized both the
water content and the hydraulic conductivity as functions of pressure head, assuming an infinite
thickness unsaturated zone.  Mishra and Neuman (2010, 2011) extended the solution of
Tartakovsky and Neuman (2007) using four parameters to represent the unsaturated zone
properties and considering a finite thickness for the unsaturated zone (Mishra and Neuman,
2010), and considered the wellbore storage as well (Mishra and Neuman, 2011). The main results
from the studies concerning vertical wells indicated that the unsaturated zone often had a major
impact on the S-shaped drawdown type curves.
A following question to ask is that are these conclusions drawn for vertical wells also
applicable for horizontal and slant wells when coupled unsaturated-saturated flow is of concern?
Specifically, how important is the wellbore orientation on groundwater flow to a horizontal or
slant well considering the coupled unsaturated-saturated flow process? In order to answer these
questions, we establish a mathematical model for groundwater flow to a general well orientation
(vertical, horizontal, and slant wells) considering the coupled unsaturated-saturated flow process.
We incorporate a three-dimensional linearized Richards' equation into a governing equation of
groundwater flow in an unconfined aquifer. We employ the Laplace-finite Fourier transform and
the method of separation of variables to solve the coupled unsaturated-saturated flow governing
equations. This paper is organized as follows, we first present the mathematical model and
solution in sections 2 and 3, respectively, then describe the results and discussion in section 4,
and summarize this study and draw conclusions in section 5.
**2. Mathematical Model**
The schematic diagrams of flow to horizontal and slant wells in an unsaturated-saturated
system are represented in Fig. 1a. and 1b, respectively. Similar to the conceptual model used by
Zhan and Zlotnik (2002), the origin of the Cartesian coordinate is located at the bottom of the
saturated zone with the $z$ axis along the upward vertical direction and the $x$ and $y$ axes along the
principal horizontal hydraulic conductivity directions. The horizontal and slant well screens are
located in the saturated zone with a distance $z_w$ from the center point of the screen $(0, 0, z_w)$ to
the bottom of the saturated zone. The slant well has three inclined angles $\gamma_x$, $\gamma_y$, and $\gamma_z$ with the
$x$, $y$, and $z$ axes, respectively, and such three angles satisfying $\cos^2(\gamma_x) + \cos^2(\gamma_y) + \cos^2(\gamma_z) =$
1. The horizontal well is a specific case of the slant well when $\gamma_z = \pi/2$. The saturated zone is
assumed as an infinite lateral extent unconfined aquifer with a slight compressibility, and is
spatially uniform and anisotropic (Tartakovsky and Neuman, 2007). The saturated zone is below
an initially horizontal water table at $z = d$, and the unsaturated zone is above $z = d$ with an
initial thickness $b$.
In order to solve the problem of groundwater flow to a horizontal or slant well, we first solve
the governing equation of groundwater flow to a point sink. The mathematical model for
groundwater flow to a point sink $(x_0, y_0, z_0)$ in a homogeneously anisotropic saturated zone is
given by
$$K_x \frac{\partial^2 s}{\partial x^2} + K_y \frac{\partial^2 s}{\partial y^2} + K_z \frac{\partial^2 s}{\partial z^2} + Q\delta(x - x_0)\delta(y - y_0)\delta(z - z_0) = S_S \frac{\partial s}{\partial t}, \quad 0 \leq z < d, \quad (1a)$$
$$s(x, y, z, 0) = 0, \qquad (1b)$$
$$\frac{\partial s}{\partial z}(x, y, z, t)|_{z=0} = 0, \qquad (1c)$$
$$\lim_{x \to \pm\infty} s(x, y, z, t) = \lim_{y \to \pm\infty} s(x, y, z, t) = 0, \qquad (1d)$$
where $s$ is the drawdown (the change in hydraulic head from the initial level) in the saturated
zone [L]; $K_x$, $K_y$ and $K_z$ are the saturated principal hydraulic conductivities in the $x$, $y$ and $z$
directions, respectively [LT$^{-1}$]; $Q$ is the pumping rate (positive for pumping and negative for
injecting) [L$^3$T$^{-1}$]; $\delta(\cdot)$ is the Dirac delta function [L$^{-1}$]; $S_S$ is the specific storage [L$^{-1}$]; $d$ is the
saturated zone thickness [L]; $t$ is time since start of pumping [T]. It is noteworthy that the aquifer
is assumed to be homogenous and spatially uniform in this study. Despite the fact that a real-
world aquifer is likely to be heterogeneous and/or non-uniform, there are evidences that a
moderately heterogeneous aquifer may sometimes behave as an averaged "homogeneous"
system for pumping-induced groundwater flow problems. This interesting phenomena may be
due to the diffusive nature of groundwater flow which can somewhat smooth out the effect of the
heterogeneity to a certain degree (Pechstein et al., 2016;Zech and Attinger, 2016).

Flow in the unsaturated zone induced by pumping in the unconfined aquifer is governed by

the Richards' equation. Due to the nonlinear nature of the Richards' equation, it is difficult to
analytically solve this equation except for some specific cases. Kroszynski and Dagan (1975)
proposed a first-order linearized unsaturated flow equation by expanding the dependent variable
in the Richards' equation as a power-function series when the pumping rate was less than $Kd^2$,
where $K$ is the saturated hydraulic conductivity of a homogeneous medium. The readers can find
the details of the linearized equation derivation in previous studies (Kroszynski and Dagan,
1975;Tartakovsky and Neuman, 2007). With such a linearized treatment, it becomes possible to
analytically solve the equation of flow in the unsaturated zone. The linearized three-dimensional
unsaturated flow equation is adopted in this study as follows,

$$k_0(z)K_x\frac{\partial^2 u}{\partial x^2} + k_0(z)K_y\frac{\partial^2 u}{\partial y^2} + K_z\frac{\partial}{\partial z}\left(k_0(z)\frac{\partial u}{\partial z}\right) = C_0(z)\frac{\partial u}{\partial t}, \quad d \leq z < d+b, \quad (2a)$$

$$u(x,y,z,0) = 0, \quad (2b)$$

$$\frac{\partial u}{\partial z}(x,y,t)|_{z=d+b} = 0, \quad (2c)$$

$$\lim_{x\to\pm\infty} u(x,y,z,t) = \lim_{y\to\pm\infty} u(x,y,z,t) = 0, \quad (2d)$$

$$k_0(z) = k(\theta_0), \quad C_0(z) = C(\theta_0), \quad (2e)$$

where $u$ is the drawdown in the unsaturated zone [L]; the functions $k_0(z)$ and $C_0(z)$ are the
zero-order approximation of the relative hydraulic conductivity [dimensionless] and the soil
moisture capacity [$L^{-1}$] at the initial water content of $\theta_0$, respectively; $k$ is the relative hydraulic
conductivity and $0 \leq k \leq 1$; $C(\geq 0)$ is the specific moisture capacity [$L^{-1}$], and $C = d\theta/d\psi$, $\theta$
is the volumetric water content [dimensionless], and $\psi$ is the pressure head [L]; $b$ is the thickness
of the unsaturated zone [L]. Similar to Tartakovsky and Neuman (2007), the unsaturated zone
properties are described with the two-parameter Gardner (1958) exponential constitutive
relationships,

$$k_0(z) = e^{\kappa(d-z)}, \quad (3a)$$

$$C_0(z) = S_y\kappa e^{\kappa(d-z)}, \quad (3b)$$

where $\kappa > 0$ is the constitutive exponent [$L^{-1}$], $S_y$ is the specific yield [dimensionless]. As
mentioned in the introduction that this two-parameter model was extended to the four-parameter
model by Mishra and Neuman (2010, 2011). The four-parameter model may be closer to the
realistic situation. However, a model with more parameters has its disadvantage as well. Firstly,
it is more difficult to determine the values of those parameters precisely from a practical
standpoint. Secondly, the predictive capability of a model with more parameters may not be
better than that of a model with less parameters. For the discussion of this issue, one may consult
the editorial messages of Voss (2011a, 2011b) and discussion by Bredehoeft (2005). In this
study, we focus on a question that how important is the wellbore orientation on groundwater flow
to a horizontal or slant well considering the coupled unsaturated-saturated flow process. To focus
on answering this question, we prefer to use a simpler model with the balance that keeping the
most important physical processes in the model but at the same time ignoring the secondary
effects.
It shows in Eq. (3b) that at the water table ($z=d$) a smaller $\kappa$ leads to a smaller $C_0(z)$ and a
larger retention capacity (Kroszynski and Dagan, 1975;Tartakovsky and Neuman, 2007), i.e.,
water in the unsaturated zone becomes more difficult to drain. In this study, we assume the upper
boundary of the unsaturated zone as a no-flow boundary condition in Eq. (2c) by neglecting the
effects of both infiltration and evaporation during the pumping. Because typical pumping tests
usually last over much shorter periods of time relative to the durations of infiltration and
evaporation processes, this assumption can hold for most field conditions, particularly for lands
with sparse vegetation where the influence of plant transpiration is limited as well.
The saturated and unsaturated flows are coupled at their interface by continuities of pressure
and vertical flux across the water table which, following linearization, take the form

$$s - u = 0, \quad z = b, \tag{4a}$$

$$\frac{\partial s}{\partial z} - \frac{\partial u}{\partial z} = 0, \quad z = b. \tag{4b}$$

Above linearized equations of (4a) and (4b) assume that the variation of water table is minor
in respect to the total saturated thickness. This assumption works better for horizontal wells and
slant wells as for vertical wells, provided that the same pumping rate is used. This is because
horizontal wells and slant wells will generate much less drawdowns over laterally broader regions;
while vertical wells tend to generate laterally more concentrated and much greater drawdown near
the pumping wells (Zhan and Zlotnik, 2002).

## 3. Solutions

### 3.1 Solution for a point sink

The solution to Eq. (1a) is obtained by the Laplace and finite cosine Fourier transform. The
Laplace domain solution of Eq. (1a) subject to initial condition Eq. (1b) and boundary conditions
Eqs. (1c) and (1d) is given as (Zhan and Zlotnik, 2002)

$$\bar{s}_D(\mathbf{r}_D, z_D, p) = \sum_{n=0}^{\infty} \frac{8\cos(\omega_n z_{0D})\cos(\omega_n z_D)}{p\Psi(\omega_n)} K_0(\Omega_n|\mathbf{r}_D - \mathbf{r}_{0D}|), \qquad (5)$$

where

$$\Omega_n = \sqrt{\omega_n^2 + p}, \ \ \Psi(\omega_n) = 2\alpha_z + \sin(2\omega_n\alpha_z)/\omega_n, \qquad (6)$$

where the subscript $D$ denotes the dimensionless terms, the definition of all dimensionless
variables are presented in the supplementary material (S1); $p$ is the Laplace transform parameter
with respect to the dimensionless time, and the overbar denotes a variable in the Laplace domain;
$\omega_n$ is the $n$-th eigenvalue of the Fourier transform, and it will be determined later; $K_0$ is the
modified second-kind Bessel function of zero-order; $\mathbf{r}_D = (x_D, y_D)$ and $\mathbf{r}_{0D} = (x_{0D}, y_{0D})$ are the
dimensionless radial vectors of the observation point and the sink point, respectively.
The solution to Eq. (2a) is obtained by the Laplace transform and the method of separation of
variables (supplementary material, S2) and is given as

$$\bar{u}_D(r_D, z_D, p) = \sum_{n=0}^{\infty} \frac{8\cos(\omega_n z_{0D})}{p\Psi(\omega_n)} K_0(\Omega_n|\mathbf{r}_D - \mathbf{r}_{0D}|) \mathcal{H}_n(z_D, p), \qquad (7)$$

where

$$\mathcal{H}_n = \begin{cases} \cos(\omega_n \alpha_z) \frac{(M+N)\exp[2N(\alpha_z+b_D)+(M-N)z_D]-(M-N)\exp[(M+N)z_D]}{(M+N)\exp[2N(\alpha_z+b_D)+(M-N)\alpha_z]-(M-N)\exp[(M+N)\alpha_z]}, & if\ \Delta > 0 \\ \cos(\omega_n \alpha_z)\exp(Mz_D-M\alpha_z)\frac{[N_1\tan(N_1(\alpha_z+b_D))-M]\sin(N_1z_D)+[M\tan(N_1(\alpha_z+b_D))+N_1]\cos(N_1z_D)}{[N_1\tan(N_1(\alpha_z+b_D))-M]\sin(N_1\alpha_z)+[M\tan(N_1(\alpha_z+b_D))+N_1]\cos(N_1\alpha_z)}, & if\ \Delta < 0 \\ \cos(\omega_n \alpha_z)\exp(Mz_D-M\alpha_z)\frac{1+M(\alpha_z+b_D)-Mz_D}{1+M(\alpha_z+b_D)-M\alpha_z}, & if\ \Delta = 0 \end{cases} \quad (8)$$

where $M = \kappa_D/2$; $N = \sqrt{\Delta}$ if $\Delta \geq 0$; $N_1 = \sqrt{-\Delta}$ if $\Delta < 0$; $\Delta = \kappa_D^2/4 + \beta p - \Omega_n^2$.
The eigenvalues of the finite cosine Fourier transform $\omega_n$ can be obtained by substituting
Eqs. (5) and (7) into the continuities of normal (vertical) flux equation (Eq. (S6b)). The detail can
be found in supplementary material (S3). On the basis of the method illustrated above, it is
straightforward to obtain the Laplace domain solutions $\bar{s}_D$ for the case of the unconfined aquifer
with a free water table boundary and without the unsaturated zone influence (Zhan and Zlotnik,
2002) (abbreviated as the ZZ solution hereinafter), and the case of the groundwater flow to a
horizontal well in an confined aquifer (Zhan et al., 2001) (abbreviated as the ZWP solution
hereinafter). The solutions $\bar{s}_D$ for these two special cases require different $\omega_n$ values. For the free
water table condition the $\omega_n$ is the root of $\omega_n \tan(\omega_n) = p/\sigma$ (Zhan and Zlotnik, 2002). For the
confined aquifer case the $\omega_n = n\pi/\alpha_z$, $n = 0,1,2,...$ (Zhan et al., 2001).

**3.2 Solution for a slant pumping well**

Due to the linearity of the mathematical models Eqs. (1) and (2), the principle of
superposition can be employed to extend the basic solutions of Eqs. (5) and (7). Thus, on the
basis of the principle of superposition, the drawdown induced by a line sink in the saturated zone
can be obtained by integrating the solution Eqs. (5) and (7) along the well axis, provided that the
pumping strength distribution along the well screen is known. Precise determination of the
pumping strength distribution along a horizontal or slant well involves complex, coupled aquifer-
pipe flow (Chen et al., 2003) in which the flow inside the wellbore (pipe flow) can experience
different stages of flow schemes from laminar, transitional turbulent, to fully developed turbulent
flow. Such complex coupled well-aquifer flow is beyond the scope of this study and one may
consult some recent studies of Blumenthal and Zhan (2016) and Wang and Zhan (2016) for more
details. However, often time one may adopt a first-order approximation of using a uniform flux
distribution to treat the horizontal or slant wells, particularly when the well screen lengths are not
extremely long (like kilometers). Such an approximation has been justified by Zhan and Zlotnik
(2002). In this study, a uniform flux distribution will be utilized for horizontal or slant wells
hereinafter to obtain the solutions.
The drawdown in saturated and unsaturated zones due to a slant pumping well can be written
as:
$$\bar{s}_{ID}(p) = \sum_{n=0}^{\infty} \frac{8\cos(\omega_n z_D)}{L_D p \Psi(\omega_n)} \int_{-\frac{L_D}{2}}^{\frac{L_D}{2}} \cos\left[\omega_n\left(z_{wD} + l\frac{\alpha_z}{\alpha_x}\cos\gamma_z\right)\right] K_0[\Omega_n F(l)]dl, \qquad (9)$$

and
$$\bar{u}_{ID}(p) = \sum_{n=0}^{\infty} \frac{8\mathcal{H}_n(z_D,p)}{L_D p \Psi(\omega_n)} \int_{-\frac{L_D}{2}}^{\frac{L_D}{2}} \cos\left[\omega_n\left(z_{wD} + l\frac{\alpha_z}{\alpha_x}\cos\gamma_z\right)\right] K_0[\Omega_n F(l)]dl, \qquad (10)$$

respectively, where $\bar{s}_{ID}$ and $\bar{u}_{ID}$ are the Laplace transforms of $s_{ID}$ and $u_{ID}$, respectively, and they
are defined in the same way as $s_D$ and $u_D$ in Eqs. (5) and (7), respectively; $L_D = \alpha_x L/d$ is the
dimensionless length of the slant well screen ($L$); $z_{wD} = \alpha_z z_w/d$ is the dimensionless elevation of
the    center    of    the    pumping    well    screen;    $l$    is    a    dummy    variable;    $F(l) =$
$\sqrt{\left(x_D - l\sin\gamma_z\cos\gamma_x\right)^2 + \left(y_D - l\frac{\alpha_y}{\alpha_x}\sin\gamma_z\cos\gamma_y\right)^2}$ . $\bar{s}_{ID}$ and $\bar{u}_{ID}$ will respectively reduce to
drawdowns in the saturated and unsaturated zones due to a horizontal well when $\gamma_z = \pi/2$. It is
noteworthy that these solutions can be straightforwardly extended to situations of location-
dependent pumping rates as long as the flux rate distribution along the wellbore is known *a priori*.
To do so, one simply modifies Eqs. (9) and (10) using a location-dependent flux function inside
the integration.
The drawdown in an observation (vertical) well located in the saturated zone that is screened
from $z_l$ to $z_u$ ($z_u > z_l$) can be calculated using the average of the point drawdown Eq. (9) along
the observation well screen (Zhan and Zlotnik, 2002):
$$\bar{s}_{oD}(p) = \sum_{n=0}^{\infty} \frac{8[\sin(\omega_n z_{uD}) - \sin(\omega_n z_{lD})]}{L_D(z_{uD} - z_{lD})\omega_n p \Psi(\omega_n)} \int_{-\frac{L_D}{2}}^{\frac{L_D}{2}} \cos\left[\omega_n\left(z_{wD} + l\frac{\alpha_z}{\alpha_x}\cos\gamma_z\right)\right] K_0[\Omega_n F(l)]dl, \qquad (11)$$
where $\bar{s}_{oD}$ is the Laplace transform of $s_{oD}$, and $s_{oD}$ is defined in the same way as $s_D$ in Eq. (5);
$z_{uD} = \alpha_z z_u/d$, $z_{lD} = \alpha_z z_l/d$.
It should be noted that our solutions do not account for the wellbore effects of the pumping
and observation wells. Indeed, the wellbore effects have introduced additional complexity to the
solutions which are already substantially more complex than the solutions excluding the
unsaturated flow process. To avoid the influence of wellbore storage effects, we make the
following proposal that could be implemented in the future investigations of coupled saturated-
unsaturated flow process: using pack systems to insulate the screens of pumping and the
observation wells, thus wellbore storages will not be a concern.
**3.3 Total volume drained from the unsaturated zone for a slant well**
The dimensionless total volume drained from the unsaturated zone to the saturated zone
(water flux across the water table) can be obtained by
$$\overline{W}_D(p) = -\int_{-\infty}^{+\infty}\int_{-\infty}^{+\infty} \frac{\partial \bar{s}_{1D}}{\partial z_D}\bigg|_{\alpha_z} dx_D dy_D = \sum_{n=0}^{\infty} \frac{16\pi \sin(\omega_n \alpha_z)\cos(\omega_n z_{wD})\sin(\omega_n \phi)}{p\Psi(\omega_n)\Omega_n^2 \phi}, \qquad (12)$$
where $\overline{W}_D$ is the Laplace transform of $W_D$, and $W_D = W\frac{4\pi\alpha_z^3}{Q}$, $W$ is the total volume drained from
the unsaturated zone; $\phi = L_D \alpha_z \cos(\gamma_z)/(2\alpha_x)$.
It is difficult to obtain closed-form solutions by analytically inverting the Laplace transforms
of Eqs. (5), (7), (9), (10) and (12) and thus a numerical inverse Laplace method is employed in
this study. There are several numerical inverse Laplace methods, such as Stehfest method
(Stehfest, 1970), Zakian method (Zakian, 1969), Fourier series method (Dubner and Abate,
1968), Talbot algorithm (Talbot, 1979), Crump technique (Crump, 1976), and de Hoog algorithm
(de Hoog et al., 1982), with each method best fitted for a particular type of problem
(Hassanzadeh and Pooladi-Darvish, 2007). Chen (1985), Zhan et al. (2009a;2009b), and Wang
and Zhan (2013) have successfully employed the Stehfest algorithm to obtain the solution in the
time domain for similar problems to this study. For references to different inverse Laplace
methods, one can consult the review of Kuhlman (2013) and Wang and Zhan (2015). In this
study we use the Stehfest method to invert the Laplace solutions into the solutions in the time
domain. In order to ensure the accuracy of the Stehfest method, several numerical exercises have
been performed against the benchmark solutions for several special cases of the investigated
problem.
## 4. Results and Discussion
**4.1 Effect of unsaturated zone parameters**
The main difference between the ZZ solution and present solution is the upper boundary
condition of the saturated zone. The ZZ solution considered linearized free surface (kinematic)
equation as the water table boundary that employed one parameter, i.e., specific yield ($S_y$) to
account for the gravity drainage after water table declining. The present solution represents
coupled water flow through both the unsaturated and saturated zones. The water table boundary
is replaced by coupled interface conditions between the unsaturated and the saturated zones.
Thus the behavior of the drawdown in the saturated zone induced by the pumping wells will be
affected by the unsaturated zone. To investigate the manner how the dimensionless constitutive
exponent $\kappa_D$ and the dimensionless unsaturated thickness $b_D$ impact the drawdown in the
saturated zone induced by a horizontal pumping well, we plot the log-log graph of $s_{1D}$ versus
$t_D/r_D^2$ (the type curves) for different $\kappa_D$ and $b_D$ in Figures 2a and 2b, respectively. We also
compare our solution to the ZZ solution (unconfined aquifer) and the ZWP solution (confined
aquifer). For convenience we assume the horizontal well screen to be situated along the $x$-
direction, i.e., $\gamma_x = 0$ and $\gamma_y = \gamma_z = \pi/2$. The other parameter values in Eq. (9) are $\sigma=1\times10^{-3}$,
$L_D=1$, $\gamma=0$, $\alpha_z=1$, $x_D=0.5$, $y_D=0.05$, $z_D=0.8$, and $z_{wD}=0.5$.

Figure 2a presents the drawdown curves in the saturated zone for different values of $\kappa_D$

($1\times10^{-5}$, $1\times10^{-3}$, $1\times10^{-1}$, $1\times10^{1}$ and $1\times10^{3}$) with a fixed dimensionless thickness of the
unsaturated zone $b_D$ of 0.5. The dimensionless constitutive exponent $\kappa_D = \kappa d/\alpha_z = \kappa d K_D^{1/3}$,
where $K_D$ is the anisotropic ratio between the vertical hydraulic conductivity and the horizontal
hydraulic conductivity.

The unsaturated flow has significant impact on drawdown curves in the saturated zone when

$\kappa_D$ is less than 10 (the unsaturated-saturated system has a large retention capacity, a small initial
saturated thickness, and/or a relatively small vertical hydraulic conductivity). The impact of
unsaturated flow decreases as $\kappa_D$ increases, becoming small or insignificant when $\kappa_D$ close to
$1\times10^3$. Our curve is almost the same as the curve of the ZZ solution when $\kappa_D = 1\times10^3$ (gray solid
curve), and gradually deviates from the ZZ solution but approaches the ZWP solution as $\kappa_D$
decreases to $1\times10^{-5}$ (black solid curve). For a fixed initial saturated thickness, when $\kappa_D$ is
smaller, i.e., the unsaturated zone has larger retention capacity and/or both the unsaturated and
saturated zones have relatively smaller vertical hydraulic conductivity, water drainage from the
unsaturated zone is impeded, forcing more water to be released from compressible storage of the
saturated zone, leading to larger drawdown in the saturated zone. The opposite is true when $\kappa_D$ is
larger. It is consistent with the findings in the vertical pumping well case (Tartakovsky and
Neuman, 2007).

It also shows in Figure 2a that the drawdown have typical "S" pattern curves while $\kappa_D \geq 0.1$.

At early times, all curves are approximately identical due to response of the confined storage and
minor effects of the upper boundary of the saturated zone; at intermediate times, the drawdowns
of the ZZ solution and our solutions increase slower than that of the ZWP solution due to
response of additional storage of the upper boundary of the saturated zone; at later times, the
drawdown increasing rates of the ZZ solution and our solutions are nearly the same as that of the
ZWP solution due to the combined effects of both storage mechanisms.

The unsaturated zone controls the effects of additional storage and upper boundary of the

saturated zone on drawdown curves. There are physical differences between the ZZ solution and
our solution. The ZZ solution uses the storage factor $S_y$ (specific yield) at upper boundary of the
saturated zone. Such a storage factor at the upper boundary is greater than the actual storage
capacity of the unsaturated zone when the unsaturated parameter $\kappa_D \leq 10$, leading to a slower
water level decline for the ZZ solution, and such effect will become insignificant for a long
pumping time. Similar to $\kappa_D$, the dimensionless unsaturated thickness $b_D$ also affects the
drawdown behavior of the saturated zone, as shown in Figure 2b for different values of $b_D$
(0.001, 0.01, 1, 10 and 100) with a fixed $\kappa_D=0.1$ and the same parameters used as Figure 2a.
Figure 2b shows that the impact of unsaturated flow increases when $b_D$ decreases. The
drawdown behavior approaches the ZWP solution when $b_D=0.001$. For large $b_D$ (=100),
however, our solution is significantly different from the ZZ solution at intermediate times
because the impact of unsaturated flow becomes significant at a fixed $\kappa_D$ of 0.1.
In order to further investigate the effects of the unsaturated zone, Figure 2c displays the
drawdown curves in the unsaturated zone ($u_{ID}$) for different values of $\kappa_D$ ($1\times10^{-5}$, $1\times10^{-3}$, $1\times10^{-1}$,
$1\times10^{1}$ and $1\times10^{3}$) at $z_D$=1.5 where the other parameters are the same as in Figure 2a. As $\kappa_D$
increases, the retention capacity of the unsaturated zone decreases, thus more water is released
from the unsaturated storage. It leads to smaller drawdown in both the unsaturated and saturated
zones. Figure 2d depicts the drawdown curves in the unsaturated zone for different values of $b_D$
(0.5, 1, 2, 10 and 100). As expected, the drawdown in the unsaturated zone decreases with  $b_D$
increasing due to the fact that more water is stored in the unsaturated zone for larger $b_D$. These
results are consistent with the findings of Mishra and Neuman (2010, 2011).
**4.2 Effect of well orientation and well screen length**
In this section, we first investigate the effect of the inclined angle of the slant well on the type
curves. Figure 3 shows the comparison between the ZZ solution and our solution with $\kappa_D = 10$
for three different angles of a slant well ($\gamma_z$= 0, $\pi/4$, and $\pi/2$) at two observation points ($z_D =$
0.9 for Figure 3a and $z_D = 0.1$ for Figure 3b) where the other parameters are the same as in
Figure 2. Obviously the smaller angle creates the larger drawdown at both observation points.
For the horizontal well ($\gamma_z = \pi/2$) the discrepancy between the ZZ solution and our solution is
larger than that for the vertical well ($\gamma_z = 0$) at upper observation point (Figure 3a). Such a
discrepancy diminishes at the lower observation point (Figure 3b). It reveals that the effects of
the unsaturated zone on the drawdown exist in any angle of inclination of a slant well for the
upper part of the aquifer, and this impact is more significant for the case of the horizontal well.
The impact of the unsaturated zone decreases when the observation point moves downward,
becoming further away from the unsaturated zone, as expected.
Here we investigate the effect of the horizontal well screen length on the drawdown.  Figure
4 illustrates the comparison between the ZZ solution and our solution for three different lengths
of well screen ($L_D$= 0.1, 1, and 10) at two observation points where the other parameters are the
same as in Figure 3. It indicates that the longer well screen leads to the smaller drawdown at both
upper and lower observation points. The discrepancy between the ZZ solution and our solution is
identical for different well screen lengths. It reveals that the effects of the unsaturated zone on
the drawdown are insensitive to the length of the horizontal well screen.
In order to clearly illustrate the drawdown pattern in the unsaturated-saturated system, the
drawdown profiles in vertical cross-sections for three different angles of a slant well ($\gamma_z$= 0, $\pi/4$,
and $\pi/2$) at different dimensionless times ($t_D$= $1\times10^3$, $1\times10^4$, and $1\times10^5$ ) are presented in Figure
5. The other parameter values in Eqs. (9) and (10) are $\sigma$=$1\times10^{-5}$, $\kappa_D$=$1\times10^3$, $L_D$=0.5, $\alpha_z$=1, $b_D$=1,
$y_D$=0.05, $z_{wD}$=0.75, $\gamma_x$ = 0, and $\gamma_y$ = $\pi/2$.  As time increases, the effect of pumping gradually
propagates into the unsaturated zone ($z_D$>1). The vertical well leads to larger drawdown in the
unsaturated zone than the slant and horizontal wells. The reason is that the vertical well screen is
closer to the unsaturated zone.
The water flux across the water table (Eq. (12)) is the volume drained from the unsaturated
zone to the saturated zone. It is somewhat related to the concept of specific yield when the
coupled unsaturated-saturated zone flow process is simplified into a saturated zone flow process
with water table served as a free upper boundary. Thus, Eq. (12) reflects the impact of the
unsaturated zone on the water flow in the saturated zone. Figure 6 shows the changes of the
dimensionless water flux across water table, $W_D$, with $t_D$ of the ZZ solution and our solution at
three angles of a slant well screen ($\gamma_z$= 0, $\pi/4$, and $\pi/2$) (Figure 6a), and at three screen lengths
of a horizontal well ($L_D$= 0.1, 1.0, and 10) (Figure 6b), where the other parameters are the same
as in Figure 3.

For early times of pumping, $W_D$ increases with time, and at the later time $W_D$ approaches an

asymptotic value that is dependent on the unsaturated parameter $\kappa_D$. $W_D$ decreases with $\kappa_D$
decreasing. The small $\kappa_D$ reflects the large retention capacity of  the unsaturated zone, and thus it
impedes water draining from the unsaturated zone during pumping. This results in more water
released from the saturated zone storage and the larger drawdown in the saturated zone (Figure
2a). The ZZ solution overestimates $W_D$ due to the fact that it neglects the effects of above
unsaturated flow (Figure 6a). The $W_D{\sim}t_D$ curves deviate from each other considerably for
different angles of a slant well, particularly at the early time. One can see from Figure 6a that $W_D$
of the vertical well ($\gamma_z$= 0) is the largest at early time, and the $W_D{\sim}t_D$ curves of three angles
eventually approach the same asymptotic value at late time. It means that the vertical well leads
to the greatest water drainage from the unsaturated zone at early time, and the effects of the well
orientation are insignificant with time increasing. Very different from the angle of a slant well,
the screen length of a horizontal well appears to have almost no impact on $W_D$ for the whole
pumping period (Figure 6b). Similar with Figure 6a, the magnitude of $W_D$ in Figure 6b is only
dependent on the unsaturated parameter $\kappa_D$.
**4.3 Synthetic pumping test**

In order to further verify our solutions and to explore the capability of our solution for

interpreting pumping test results in the unsaturated-saturated system, we have conducted a
synthetic numerical simulation. The synthetic case considers a pumping test in an unconfined
aquifer with a slant pumping well ($\gamma_z=\pi/4$, $\gamma_x=0$, and $\gamma_y= \pi/2$). The aquifer parameter values
are as follows. The unconfined aquifer thickness $d$ is10 m, the above unsaturated zone thickness
$b$ is 5 m, the horizontal conductivity $K_x = K_y$=0.06 m/min, the vertical conductivity $K_z$=0.5$K_x$,
the specific storage $S_S$=1×10$^{-4}$ m$^{-1}$, and the specific yield $S_y$=0.3. The unsaturated flow is
described by Eqs. (2) and (3) with the constitutive exponent $\kappa$= 0.1 m$^{-1}$. The discharge rate of the
pumping well $Q$=1 m$^3$/min, the length of the pumping well screen $L$ is 5 m, and the center of
well screen locates at ($x$=0, $y$=0, $z$=5 m).

The coupled equations (1) -(4) of the unsaturated-saturated system are numerically solved by

COMSOL Multiphysics, a robust Galerkin finite-element software package that includes a partial
differential equation (PDE) solver for modeling the type of governing equations of this study.
Fig. 7a shows the spatial discretization of our COMSOL model, in which tetrahedrons are used
as elements for the three-dimensional model, and the elements near both the pumping well and
the unsaturated-saturated interface are refined. The number of tetrahedral elements is 328358.
The time step increases exponentially, and the total number of time steps is 100, with a total
simulation time of 220 min. Fig. 7b presents an example for the vertical profiles (the $xz$-plane) of
the drawdown in the unsaturated-saturated system at $t$=210 min. Fig. 7b indicates that the
COMSOL model well reproduces the drawdown in the unsaturated-saturated system induced by
a slant pumping well.

Firstly, we verify our solutions by comparing the drawdowns in both the saturated and

unsaturated zones with the numerical solution for the same aquifer parameter values. Figs. 8a
and 8b show the drawdown curves in the saturated zone at an observation point of ($x$=0, $y$=1 m,
$z$=9 m) and the drawdown curves in the unsaturated zone at an observation point of ($x$=0, $y$=1 m,
$z$=11 m), respectively, using the numerical solution (triangles) and our solution (solid curves).
These figures indicate that in general our solution satisfactorily fits the numerical solution in
both the saturated and unsaturated zones, although the agreement becomes less satisfactorily (but
acceptable) at late times. The sizes of the tetrahedral elements will affect the accuracy of the
numerical solution, especially near the pumping well and the unsaturated-saturated interface.
Although we refine the mesh at these places, the sizes of these elements may be insufficiently
small to completely remove the numerical errors near those places. Our numerical exercises
show that a finer element discretization for this model leads to substantially greater
computational cost, probably due to the three-dimensional nature of the model.

Secondly, we investigate the errors for using the ZWP and ZZ solutions to explain the

drawdown curves in the unsaturated-saturated system induced by the slant pumping well. Fig. 8a
shows a least squares fit of the ZWP (dashed curves) and ZZ (dotted curves) solutions to the
numerical solution, yielding parameter estimates $K_x = K_y$=0.13 m/min, $S_S$=1.1$\times$10$^{-2}$ m$^{-1}$ (for
the ZWP solution), and $K_x = K_y$=0.03 m/min, $S_S$=2.3$\times$10$^{-4}$ m$^{-1}$, and $S_y$=0.32 (for the ZZ
solution), respectively. Obviously, the ZWP solution fails to fit the numerical solution entirely
and significantly overestimates the horizontal hydraulic conductivity and the specific storage
with one or two orders of magnitude due to the fact that it is a confined-aquifer solution. The ZZ
solution dramatically deviates from the numerical solution at the early and intermediate times
and it agrees with the numerical solution at late time. The ZZ solution underestimates the
horizontal hydraulic conductivity and overestimates the specific storage and the specific yield.

A major disadvantage of the two older models (the ZWP and ZZ models) is that they do not

consider the unsaturated flow process, thus they cannot be used to characterize the parameters of
the unsaturated zone. The newer model developed in this study, however, is capable of
characterizing parameters of both the saturated and unsaturated zones. As far as we know, this
represents a significant improvement over the older models. Furthermore, as the older models do
not consider the unsaturated flow process proven to be important for producing the drawdown-
time curves in the saturated zone, they often cannot satisfactorily reproduce the observed
drawdown-time curves in the saturated zone in actual real-world aquifer pumping tests. The
newer model has resolved this issue successfully because the used conceptual model is closer to
the physical reality of flow in an unsaturated-saturated system.

## 448    5. Summary and Conclusions

The coupled unsaturated-saturated flow process induced by vertical, horizontal, and slant
pumping wells is investigated in this study. A mathematical model for such a coupled
unsaturated-saturated flow process is presented. The flow in the saturated zone is described by a
three-dimensional governing equation, and the flow in the unsaturated zone is described by a
three-dimensional Richards' equation. The unsaturated zone properties are represented by the
Gardner (1958) exponential relationships. The Laplace domain solutions are derived using
Laplace transform and the method of separation of variables, and the time domain solutions are
obtained using the Stehfest method (Stehfest, 1970). The solution is compared with the solutions
proposed by Zhan et al. (2001) (confined aquifer, the ZWP solution) and Zhan and Zlotnik
(2002) (unconfined aquifer, the ZZ solution) and is verified using the finite-element numerical
solution. The conclusions of this study can be summarized as follows:
1)  The unsaturated flow has significant impact on drawdown in unconfined aquifers induced by

the horizontal pumping well when dimensionless constitutive exponent $\kappa_D$ is less than 10 (the

large retention capacity of the unsaturated zone, the small initial saturated thickness, and/or the

small vertical hydraulic conductivity). For the large $\kappa_D$ ($=1 \times 10^3$), the drawdown curves

approach the solution of the unconfined aquifer with the linearized free water table boundary

(the ZZ solution). For the small $\kappa_D$ ($= 1 \times 10^{-5}$ ), the drawdown curves approach the solution

of the confined aquifer (the ZWP solution).
2) For the small dimensionless unsaturated thickness $b_D (= 0.001)$, the drawdown curves
approach the ZWP solution. For the large unsaturated thickness $b_D (= 100)$, the drawdown
curves do not approach the ZZ solution because the impact of the unsaturated flow becomes
significant at a fixed $\kappa_D$ of 0.1.
3) The effects of the unsaturated zone on the drawdown exist in any angle of inclination of a slant
well, and this impact is more significant for the case of the horizontal well. The effects of the
unsaturated zone on the drawdown are insensitive to the length of the horizontal well screen.
4) For the early time of pumping, the water volume drained from the unsaturated zone ($W$) to the
saturated zone increases with time, and with time progressing, $W$ approaches an asymptotic
value that is dependent on the unsaturated parameter $\kappa_D$. The vertical well leads to the largest
$W$ value during the early time of pumping, and the effects of the well orientation become
insignificant at the late time. The screen length of the horizontal well does not affect $W$ for the
whole pumping period.
5) By comparison with synthetic pumping test data generated by the finite-element numerical
model of COMSOL, one can see that our solution well reproduces the drawdown curves in
both the saturated and unsaturated zones while both the ZWP and ZZ solutions fail to fit the
drawdown curves and they either underestimate or overestimate the horizontal hydraulic
conductivity, the specific storage, and the specific yield.

**Acknowledgement**

This study was partially supported with the research grants from the National Nature Science Foundation of China (41330314, 41272260, 41302180, 41521001, 41372253), the National Key project ''Water Pollution Control'' of China (2015ZX07204-007), and the Natural Science Foundation of Jiangsu Province (BK20130571). We thank Dr. Shlomo P. Neuman and another anonymous reviewer for their constructive comments for us to revise the manuscript.

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

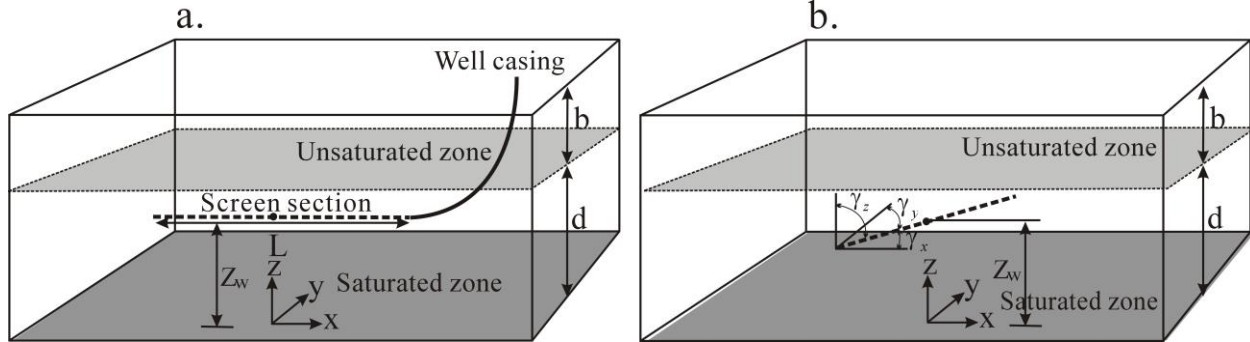


**Figure 1** The schematic diagram of groundwater flow to a horizontal well (a) and a slant well (b) in an
unsaturated-saturated system.


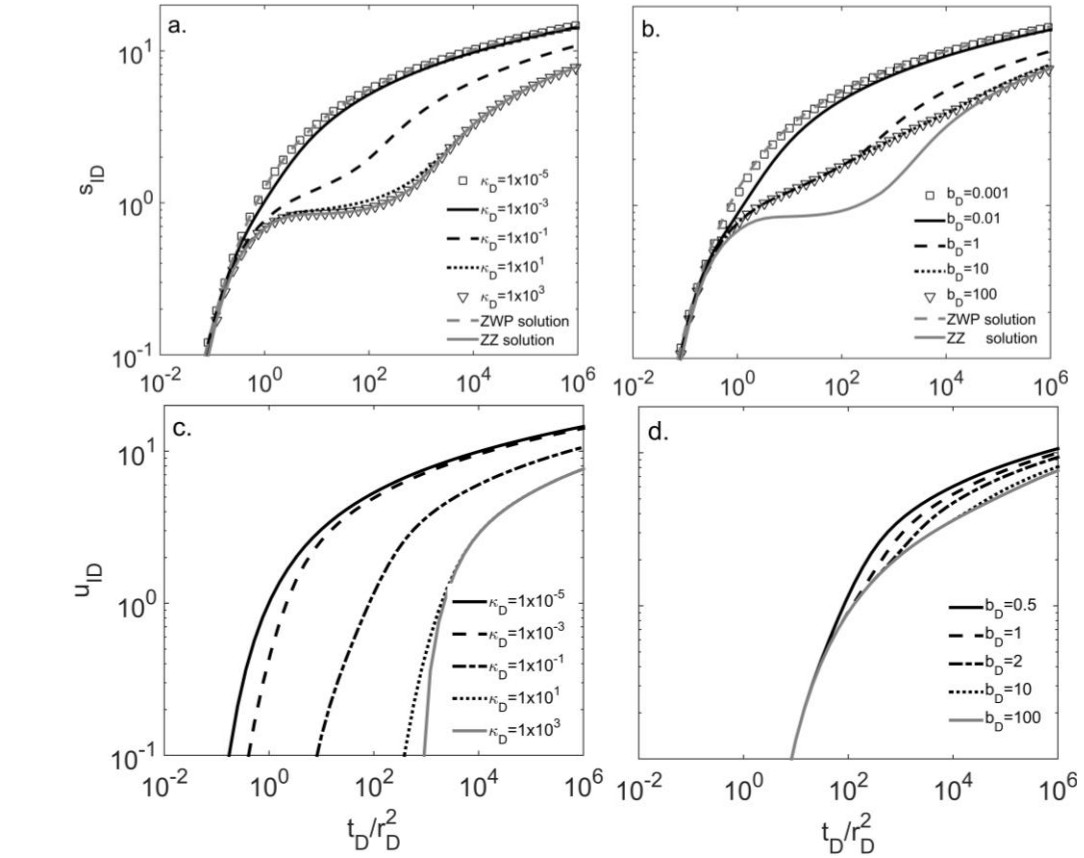


**Figure 2** a) log-log plot of $s_{ID}$ against $t_D/r_D^2$ for different values of the dimensionless unsaturated parameter $\kappa_D$, the ZWP solution (confined aquifer) and the ZZ solution (unconfined aquifer), b) log-log plot of $s_{ID}$ against $t_D/r_D^2$ for different values of the dimensionless unsaturated thickness $b_D$, the ZWP solution (confined aquifer) and the ZZ solution (unconfined aquifer), c) log-log plot of $u_{ID}$ against $t_D/r_D^2$ for different values of the dimensionless unsaturated parameter $\kappa_D$, and d) log-log plot of $u_{ID}$ against $t_D/r_D^2$ for different values of the dimensionless unsaturated thickness $b_D$.


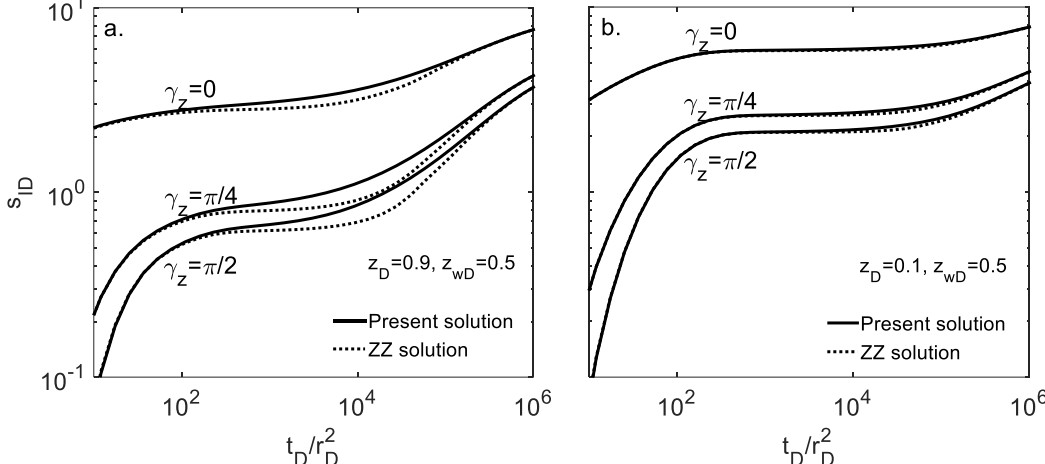

**Figure 3** log-log plot of $s_{ID}$ against $t_D/r_D^2$ for different angles of well screen and comparison with the ZZ
solution for a) dimensionless piezometer location (0, 0.05, 0.9), and b) dimensionless piezometer location
626 (0, 0.05, 0.1).


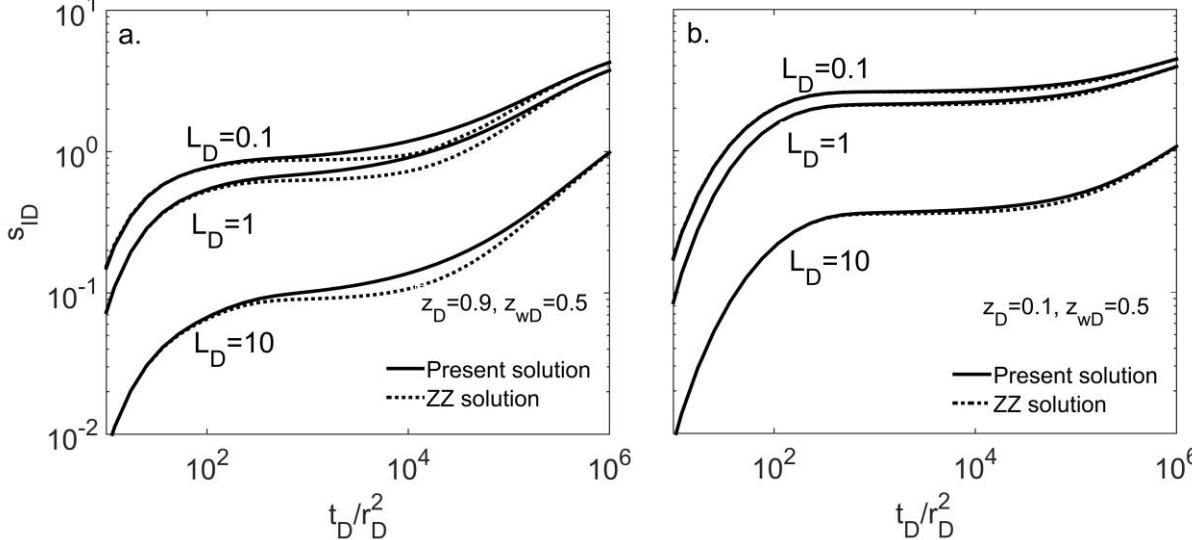


**Figure 4** log-log plot of $s_{ID}$ against $t_D/r_D^2$ for different dimensionless lengths of horizontal well screen
and comparison with the ZZ solution for a) dimensionless piezometer location (0, 0.05, 0.9), and b)
dimensionless piezometer location (0, 0.05, 0.1).



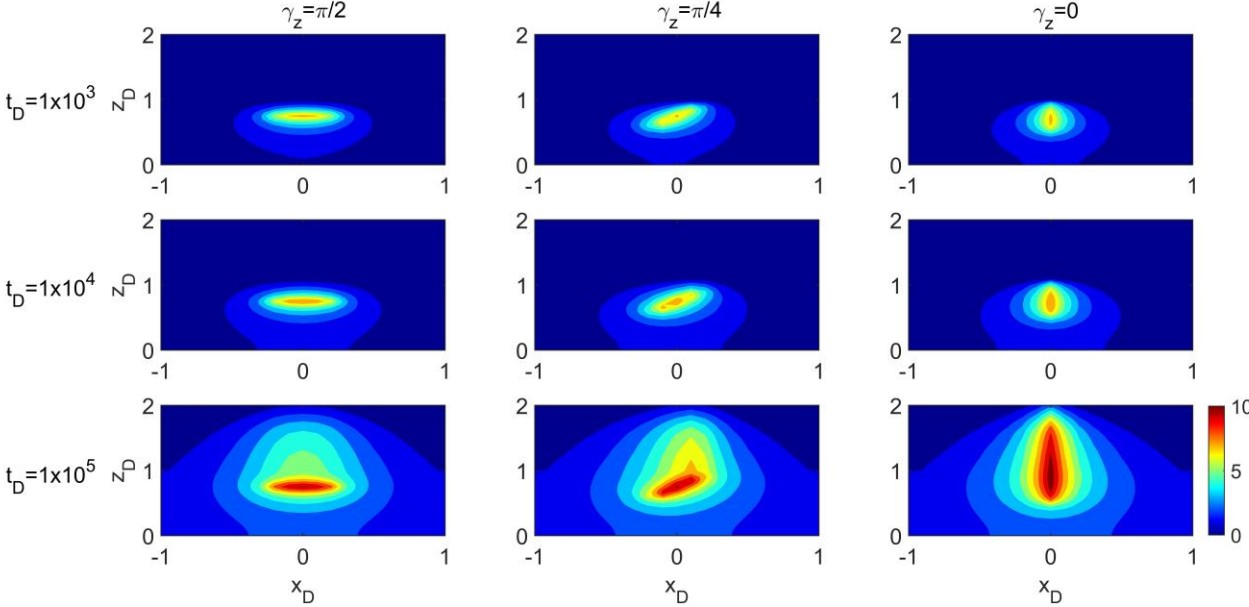


**Figure 5** Vertical profiles of $s_{ID}$ in saturated and $u_{ID}$ in unsaturated zones for different angles of well
screen corresponding to various dimensionless times.

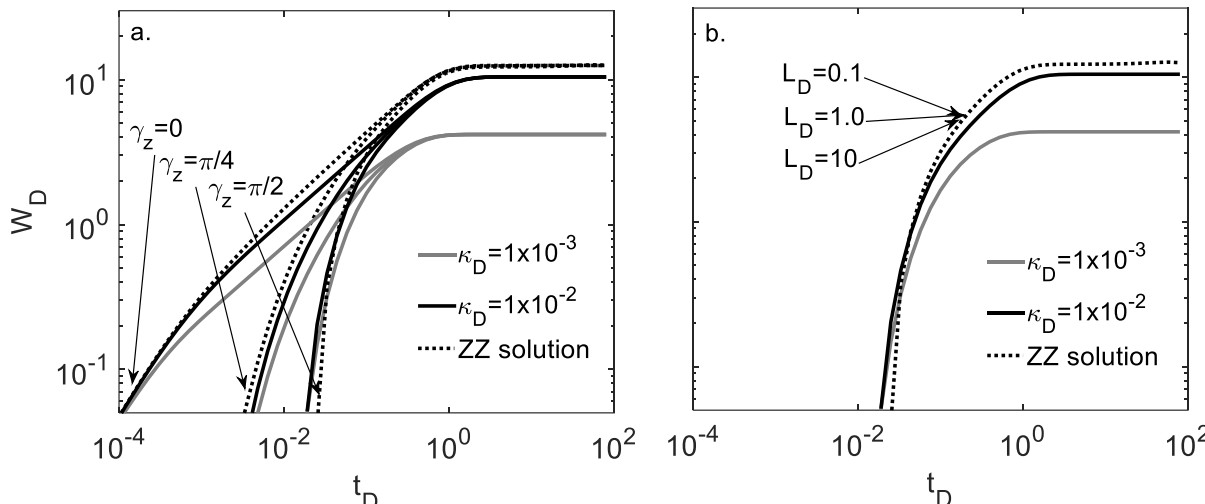


**Figure 6** log-log plot of $W_D$ against $t_D$ for different values of the dimensionless unsaturated parameter
$\kappa_D$ and the ZZ solution with a) three angles of the slant well screen ($\gamma_z = 0$, $\pi/4$, and $\pi/2$), and b) three
dimensionless lengths of the horizontal well screen ($L_D = 0.1$, 1.0, and 10).



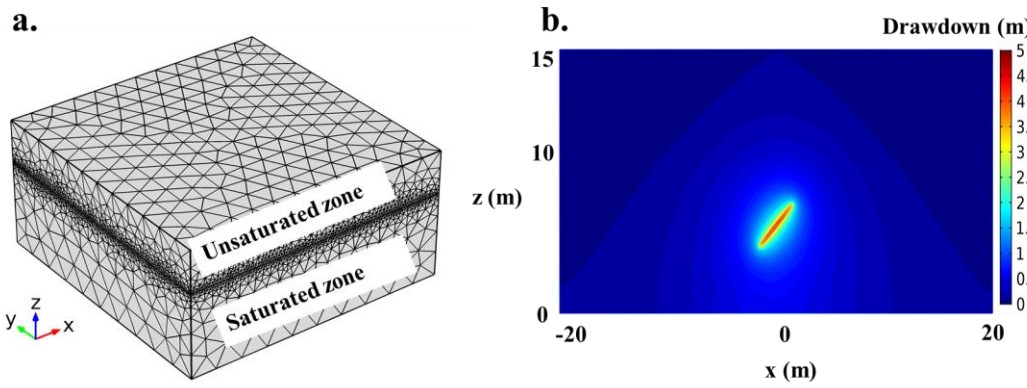


**Figure 7** a) The grid mesh of the unsaturated-saturated system used in the Galerkin finite element
COMSOL Multiphasic program, and b) the vertical profiles (*xz*-planes) of the drawdown in the
unsaturated-saturated system on *t*=210 min for the synthetic case.

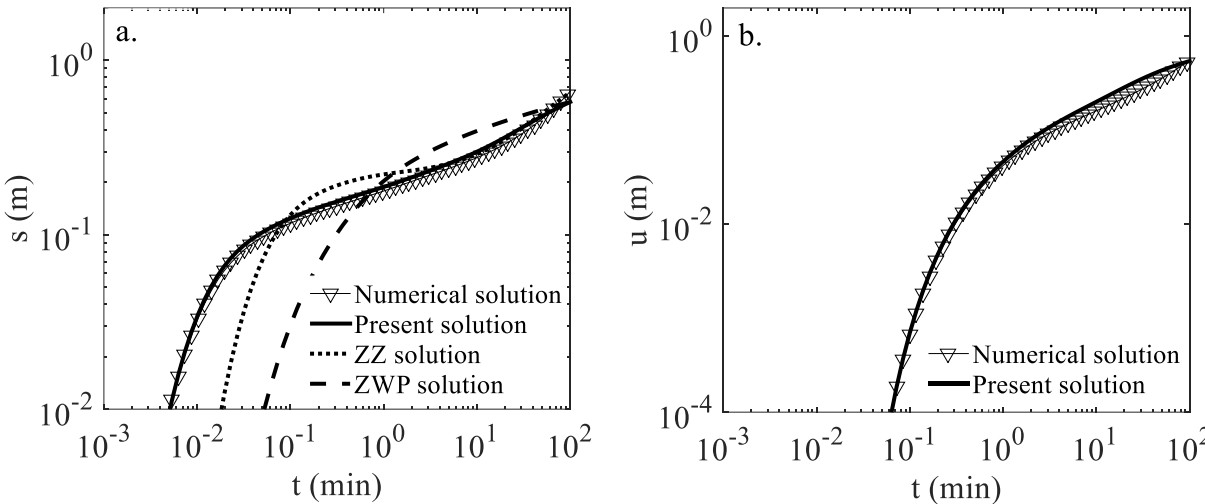


**Figure 8** a) Comparison of synthetic drawdown in saturated zone generating from numerical solution
with fitted analytical solutions using ZZ solution, ZWP solution and our solution, and b) Comparison of
synthetic drawdown in unsaturated zone generating from numerical solution with our solution.