# Peer review of "On Coupled Unsaturated-Saturated Flow Process Induced by Vertical,"

_Hydrology and Earth System Sciences, 2016_

## Referee Comment (RC1) · Anonymous Referee #1 · 7 Nov 2016

**General comments**

In this manuscript, the authors develop the model for the interpretation of pumping test in an aquifer with variable saturation for both horizontal as well as partially inclined wells. The model is derived using a semi-analytical solution of the coupled saturated-unsaturated flow processes. To facilitate an analytical treatment of the non-linear Richards equation, the authors linearize the equations by assuming low pumping rates. Both the saturated and unsaturated systems is coupled by assuming continuity of pressure and fluxes at the interface. The semi-analytical solution of this coupled system is eventually used to infer the hydraulic parameters of the subsurface.

The manuscript itself is well structured but poorly written. A significant revision of the English is needed. The introduction gives an adequate overview on the relevant questions and properly motivates the study. The methods section provides the reader with the necessary information on the mathematical background with more information being provided in the Supplementary Information. The results are presented twofold. The analytical solutions to the coupled systems are first derived for a number of special cases. Numerical solutions for these cases are then presented and discussed in Section 4; Results and discussion. These numerical results are given in a way that it easy to follow and understand. The data given through figures clear and sufficient to support the conclusions drawn by the authors. The presented conclusions may be relevant for the Scientific Community interested in horizontal well drilling. However, the authors fails to properly motivate the need for their study and to present results that are relevant to practitioners. In conclusion, I think the manuscript needs major revisions before being eligible for publication in HESS.

In the following, I will list a number my concerns that should be fixed to improve the manuscript.

**Major Concerns**

- The used geometry and the considered processes were chosen to be very simple in order to facilitate the use of analytical tools for the investigation. For example, the geometry is considered to be spatially uniform and exhibits no anisotropy which is unusual for a three-dimensional medium. Such simple models aren't bad if the insight derived from them can be properly transferred to real-world problems. At this point, the authors need to explain why they think these simplifications are possible and critically assess their impact on their results.

- In my opinion, the authors fail to properly put their results into context. Instead

the authors should better demonstrate why and when the difference between their model and two older models for pumping in coupled saturated-unsaturated systems matter. I am not doubting that their approach has not been done before, but this does not automatically make it relevant and interesting. To show that, the authors should begin by explaining some problems of the two older models, and subsequently demonstrate how their newer approach can remedy these problems. In particular, they should be able to discuss how these observed differences relate to actual real-world features.

**Minor Concerns**

- There are consistently no captions for the Figures.

- Furthermore, the quality of the figures is generally bad. The authors may want to use another compression format or a lower compression rate.

- Line 197: I think the authors mean that the linearity of the system allows to superimpose the solutions of Equations (5) and (7).

- Line 202: The authors mention turbulent flow. How is this possible for a linear system?

- Line 207: The authors use a uniform flux rate for the spatially extended wells. Can the approach also used with arbitrary flux rates?

- Line 238: Here the authors say that the Stehfest algorithm was sufficiently accurate for the flow problem. That is just an assertion and should be backed up by at least some evidence.

[Figure]

- Line 242: The use of the word 'real-time solutions' is confusing here. I first thought the authors would derive the solution on the fly. Maybe they should say 'solution in the time domain'.

- Line 242-244: This sentence is confusion. Please reformulate.

- Line 248: What is the kinematic equation? I am not familiar with this approach.

- Line 256: For most of the time the authors use the passive voice in the manuscript. Here they suddenly switch into the active voice. Although I like the active voice much better, the authors should be consistent.

- Line 266: The gray line mentioned here is actually hard to see, due to the afore-mentioned bad quality of the figures.

- Line 270-273: This sentence is long and confusing. Please consider to reformulate this statement.

- Line 280: The authors use the passive voice with respect to a figure. This is confusion and, to the best of my knowledge, not proper English.

- Line 318: This is not a proper sentence. Please reformulate.

**Typos**

As mentioned above, the manuscript suffers from poor spelling, grammar and several typos. In the following, I will provide a short list of examples.

- Line 69: of *an* unsaturated

- Line 105: with a *slightly*

- Line 153: much *shorter* periods

- Line 155: where the influence of plant transpiration is

- Line 176: *with* respect to

- Line 176: overbar *denotes*

- Line 232: and thus *a* numerical

- Line 254: the manner *how* the

- Line 259: For convenience

- Line 259 well screen *to be situated* along

- Line 285: at *later times*

- Line 297: For large

- Line 314: to *a* smaller

- Line 325: *closer* to

- Line 326: across *the* water

- Line 329: the *impact* of

- Line 335 For early times

- Line 338: *This* results

- Line 365: of *the* unsaturated

---

## Referee Comment (RC2) · S. P. Neuman (Referee) · 11 Nov 2016

In this brief manuscript, the authors (LZZL) present a new semi-analytical solution for flow to a slanted well (including horizontal and vertical) of zero radius in a uniform anisotropic unconfined aquifer considering unsaturated flow above the water table. Details of the solution are included in supplementary material and appear to be correct. The underlying conceptual-mathematical model is similar to that of Tartakovsky and Neuman (2007) for a vertical well, with the exception that LZZL take the unsaturated zone to be finite (TN took it to be infinite). Their method of solution is somewhat different from that of TN. The authors evaluate their solution numerically and present it in the forms of time-drawdown curves in the saturated zone, synoptic profiles of dimensionless drawdown in both the saturated and the unsaturated zones, and flow rate across the water table, for a range of dimensionless parameters, concluding that the unsaturated zone has a significant effect on system behavior in all cases.

I find the paper to be clearly written and the mathematics well explained. I do, however, have a few fundamental questions to the authors:
1. LZZL are aware that the conceptual-mathematical model of TN has been superseded by a more general model due to Mishra and Neuman (2010, 2011). In addition to having rendered the unsaturated zone finite (as do LZZL), the most important extensions introduced by MN are representations of unsaturated material properties by 4 (instead of 2) parameters, in a manner similar to that of Mathias and Butler (2006); accounting for storage in the pumping well (rather than treating this well as a line sink); and accounting for delayed response of piezometers and observation wells due to storage in these devices. TN have demonstrated that the four-parameter representation leads to more realistic estimates of aquifer parameters, based on observed drawdowns, than does the two-parameter model. They also demonstrated that storage of water in pumping and observation wells have significant impacts on drawdowns below and above the water table. My question to LZZL: Why have you not worked with a four-parameter model, and why have you not accounted for pumping and observation well storage?
2. MN have demonstrated that the unsaturated zone may or may not have a significant impact on drawdown below the water table depending on the choice of system parameters and mode of observation. My question to LZZL: What justifies your blanket statement that the unsaturated zone has a significant effect on system behavior, without any qualifications?
3. MN have shown time-drawdown curves for the unsaturated zone; why have LZZL not done likewise? New developments in unsaturated zone sensor technology will likely make it practical, in the not too distant future, to observe unsaturated zone behavior at depth and use the MN (or LZZL) solution to interpret such observations quantitatively.
4. MN have used their solution to analyze published pumping test results, demonstrating (as already noted) that their new model provides more realistic parameter estimates than did earlier models, including that of TN (and hence, I conclude, LZZL). Why have LZZL not done the same?
5. MN have shown that their solution allows estimating unsaturated zone properties based on observed drawdowns in the saturated zone. Why have LZZL not attempted to do the same?

In summary, the LZZL solution is interesting in that it is the first to consider slanted wells in the unconfined aquifer context. It however rests on a somewhat limited and outdated conceptualmathematical model of the system; would it be possible for the authors to remedy this? In any case, the authors should show graphically how the unsaturated zone responds to pumping and, if at all possible, use their model to analyze real (or at the least synthetically generated) pumping test data.

Review by Shlomo P. Neuman

---

## Author Comment (AC1) · 8 Jan 2017

**Responses to the Comments by Reviewers**

We thank Professor Neuman and an anonymous reviewer for their constructive comments. The manuscript has been significantly improved by addressing the comments. The following are our point-to-point responses to their comments.

**Responses to the Comments from Reviewer #1**

**General comments**
*In this manuscript, the authors develop the model for the interpretation of pumping test in an aquifer with variable saturation for both horizontal as well as partially inclined wells. The model is derived using a semi-analytical solution of the coupled saturated-unsaturated flow processes. To facilitate an analytical treatment of the nonlinear Richards equation, the authors linearize the equations by assuming low pumping rates. Both the saturated and unsaturated systems is coupled by assuming continuity of pressure and fluxes at the interface. The semi-analytical solution of this coupled system is eventually used to infer the hydraulic parameters of the subsurface.*
*The manuscript itself is well structured but poorly written. A significant revision of the English is needed. The introduction gives an adequate overview on the relevant questions and properly motivates the study. The methods section provides the reader with the necessary information on the mathematical background with more information being provided in the Supplementary Information. The results are presented twofold. The analytical solutions to the coupled systems are first derived for a number of special cases. Numerical solutions for these cases are then presented and discussed in Section 4; Results and discussion. These numerical results are given in a way that it easy to follow and understand. The data given through figures clear and sufficient to support the conclusions drawn by the authors. The presented conclusions may be relevant for the Scientific Community interested in horizontal well drilling. However, the authors fail to properly motivate the need for their study and to present results that are relevant to practitioners. In conclusion, I think the manuscript needs major revisions before being eligible for publication in HESS.*

**Response:** Thank you for the positive comment. We have improved the English of the revised manuscript as possible as we can. Two co-authors of the manuscript, Drs. Zhan and Zhang both have more than 20 years teaching & research experiences in US universities (Texas A&M Univ. and Univ. of Iowa), and they have carefully checked the English of the manuscript. In terms of the practical relevance, this revised manuscript includes a new section that provides a robust and accurate method for simultaneous determination of parameters of both saturated zone (hydraulic conductivity and specific storage) and unsaturated zone (specific yield and the constitutive exponent of the Gardner exponential model). For details, please refer to subsection 4.3 and Figure 8. We believe this revised manuscript meets the requirement of HESS and is now ready for publication.

**Major Concerns**
*1. The used geometry and the considered processes were chosen to be very simple in order to facilitate the use of analytical tools for the investigation. For example, the geometry is considered to be spatially uniform and exhibits no anisotropy which is unusual for a three-dimensional medium. Such simple models aren't bad if the insight derived from them can be properly transferred to real-world problems. At this point, the authors need to explain why they think these simplifications are possible and critically assess their impact on their results.*

**Response:** Actually our model (Eqs. (1) and (2)) have considered the anisotropy of the aquifer. For the purpose of mathematical convenience, the analytical solutions of Eqs. (5) and (7) are written as dimensionless form in which the $K_x$, $K_y$ and $K_z$ terms are lumped into dimensionless variables $x_D$, $y_D$ and $z_D$. One can easily see the impacts of anisotropy by converting the final results from dimensionless formats into dimensional formats.

In this study, we assume that the aquifer is homogenous and spatially uniform, an assumption that is almost universally adopted in previous studies about the analytical models of well hydraulics. Despite of its idealization, such analytical models are still indispensable and relevant to real-world problems, probably because of the following reasons.

Firstly, analytical models with idealized homogenous and spatially uniform assumption provide benchmark solutions which are required to test any numerical models before their usage for simulating a heterogeneous, non-uniform real-world aquifer. Without such benchmark solutions, there is almost no-way to tell if the numerical models are valid or not.

Secondly, despite that a complex numerical model can deal with a heterogeneous, non-uniform real-world aquifer, it usually needs more parameters which are often not available or not known to sufficient accuracy. Under such a circumstance, a simple analytical model may be an alternative for providing a baseline assessment of the flow system.

Thirdly, although a real-world aquifer is likely to be heterogeneous and/or non-uniform, there are evidences that a moderately heterogeneous aquifer may sometimes behave as an averaged "homogeneous" system for pumping-induced groundwater flow problems. This interesting phenomena may be due to the diffusive nature of groundwater flow which can somewhat smooth out the effect of the heterogeneity to a certain degree (Pechstein et al., 2016;Zech and Attinger, 2016). Please see lines 122-128 of the revised text.

*2. In my opinion, the authors fail to properly put their results into context. Instead the authors should better demonstrate why and when the difference between their model and two older models for pumping in coupled saturated-unsaturated systems matter. I am not doubting that their approach has not been done before, but this does not automatically make it relevant and interesting. To show that, the authors should begin by explaining some problems of the two older models, and subsequently demonstrate how their newer approach can remedy these problems. In particular, they should be able to discuss how these observed differences relate to actual real-world features.*

**Response:** Implemented. We adopted this comment and added subsection 4.3 and Figure 8 to demonstrate the problems of two older models (ZWP and ZZ models) on explaining the drawdown curves in the saturated-unsaturated system. A major disadvantage of the two older models (ZWP and ZZ models) is that they did not consider the unsaturated flow process, thus they cannot be used to characterize the parameters of the unsaturated zone. The newer model developed in this study, however, is capable of characterizing parameters of both the saturated and unsaturated zones. As far as we know, this represents a significant improvement over the older models. Furthermore, as the older models did not consider the unsaturated flow process that was proven to be important for producing the drawdown-time curves in the saturated zone, they often cannot satisfactorily reproduce the observed drawdown-time curves in the saturated zone in actual real-world aquifer pumping tests. The newer model has resolved this issue successfully. Please see lines 441-450 of the revised text for addressing this point.

To further demonstrate the difference of the older and newer models, we have conducted a numerical simulation using the finite-element model COMSOL for a synthetic case that considers the unsaturated flow process for a slant well pumping test in an unconfined aquifer. The results show that our solution nicely reproduces the drawdown curves in both the saturated and unsaturated zones; while the ZWP and ZZ solutions fail to fit the drawdown curves and they either underestimate or overestimate horizontal hydraulic conductivity, specific storage and specific yields due to its lack of consideration of the effect of unsaturated zone. Please see the details in the subsection 4.3 and Figure 8.

**Minor Concerns**
*• There are consistently no captions for the Figures.*

**Response:** Implemented. In the previous version, we included the figure captions at the end of the text. Now we added the proper caption below each Figure in the revised manuscript.

*• Furthermore, the quality of the figures is generally bad. The authors may want to use another compression format or a lower compression rate.*

**Response:** Implemented. We have reproduced all figures using higher resolution (600 dpi). All figures are more clear now.

*• Line 197: I think the authors mean that the linearity of the system allows to superimpose the solutions of Equations (5) and (7).*

**Response:** Implemented. We added this statement on lines 215-216.

*• Line 202: The authors mention turbulent flow. How is this possible for a linear system?*

**Response:** The point to mention turbulent flow is to justify the use of a uniform flux distribution along the screened section of the pumping well. This assumption may not hold for a very long horizontal well in which complex well-aquifer flow may be a concern. When dealing with pumping using a very long horizontal well, flow inside the wellbore may not be neglected. Such in-well flow (called pipe flow by some investigators) could be considerably different from Darcian flow in the aquifer as it may experience various flow schemes such as laminar and/or turbulent flow, depending on the Reynolds number and friction coefficient inside the wellbore.

Such complex well-aquifer flow is beyond the scope of this manuscript and one may consult some recent studies of Wang and Zhan (2016) and Blumenthal and Zhan (2016) for more details. Nevertheless, we may incorporate such coupled well-aquifer flow into the model of this manuscript in a future study. For more details, please see lines 223-225.

*• Line 207: The authors use a uniform flux rate for the spatially extended wells. Can the approach also used with arbitrary flux rates?*

**Response:** Implemented. The present solutions (Eqs. (9) and (10)) used a uniform flux rate for the spatially extended well. These solutions can be straightforwardly extended to situations of arbitrary flux rates as long as the flux rate distribution along the wellbore is known *a priori*. To do so, one simply modifies Eqs. (9) and (10) using a location-dependent flux function inside the integration there. Please see lines 240-244 of the revised text for addressing this point.

*• Line 238: Here the authors say that the Stehfest algorithm was sufficiently accurate for the flow problem. That is just an assertion and should be backed up by at least some evidence.*

**Response:** Implemented. We deleted the 'the Stehfest algorithm was sufficiently accurate for the flow problem' as it appears to be too assertive. However, the Stehfest method was successfully employed by many researchers to solve the problems very similar (but not exactly) to this study, e.g., Chen (1985), Zhan et al. (2009a;2009b), and Wang and Zhan (2013). Thus we are confident that the results are robust and accurate. Another piece of evidence of the reliability of the solution is that it matches very well with a high-resolution numerical simulation using a finite-element method of COMSOL. Please see lines 416-429 for details.

*• Line 242: The use of the word 'real-time solutions' is confusing here. I first thought the authors would derive the solution on the fly. Maybe they should say 'solution in the time domain'.*

**Response:** Implemented. We replaced 'real-time solutions' with 'solution in the time domain' on lines 271 and 272.

*• Line 242-244: This sentence is confusion. Please reformulate.*

**Response:** Implemented. We replaced this sentence with "In order to ensure the accuracy of the Stehfest method, several numerical exercises have been performed by comparing with the benchmark solutions for several special cases of the investigated problem" on lines 275-277.

• *Line 248: What is the kinematic equation? I am not familiar with this approach.*

**Response:** Implemented. It is free surface equation of the unconfined aquifer. We revised it as "linearized free surface (kinematic) equation" on line 281.

• Line 256: For most of the time the authors use the passive voice in the manuscript. Here they suddenly switch into the active voice. Although I like the active voice much better, the authors should be consistent.

**Response:** This is a valuable comment. However, after a careful consideration, we prefer to keep the use of passive and active voices as is, and our reasons are as follows. When discussing the results and conclusions, we prefer to use active voices, emphasizing that those are our interpretation. For the rest parts of the paper, we prefer to use passive voice, emphasizing that those are objective statements of facts.

• Line 266: The gray line mentioned here is actually hard to see, due to the aforementioned bad quality of the figures.

**Response:** Implemented. We revised the quality of the figures.

• Line 270-273: This sentence is long and confusing. Please consider to reformulate this statement.

**Response:** Implemented. We revised this sentence on lines 297-306.

• Line 280: The authors use the passive voice with respect to a figure. This is confusion and, to the best of my knowledge, not proper English.

**Response:** It was the active voice.

• Line 318: This is not a proper sentence. Please reformulate.

**Response:** Implemented.

**Typos**
As mentioned above, the manuscript suffers from poor spelling, grammar and several typos. In the following, I will provide a short list of examples.

**Response:** Implemented. We improved the writing of the revision manuscript as possible as we can. Please also see our reply to the general comment of this reviewer.

• Line 69: of an unsaturated

**Response:** Implemented.

• Line 105: with a slightly

**Response:** Implemented.

• Line 153: much shorter periods

**Response:** Implemented.

• Line 155: where the influence of plant transpiration is

**Response:** Implemented.

• Line 176: with respect to

**Response:** Implemented.

• Line 176: overbar denotes

**Response:** Implemented.

• Line 232: and thus a numerical

**Response:** Implemented.

• Line 254: the manner how the
**Response:** Implemented.

• Line 259: For convenience

**Response:** Implemented.

• Line 259 well screen to be situated along

**Response:** Implemented.

• Line 285: at later times

**Response:** Implemented.

• Line 297: For large

**Response:** Implemented.

• Line 314: to a smaller

**Response:** Implemented.

• Line 325: closer to

**Response:** Implemented.

• Line 326: across the water

**Response:** Implemented.

• Line 329: the impact of

**Response:** Implemented.

• Line 335 For early times

**Response:** Implemented.

• Line 338: This results

**Response:** Implemented.

• Line 365: of the unsaturated

**Response:** Implemented.

**Responses to the Comments from Professor Neuman (Reviewer #2)**

**General comments**

 In this brief manuscript, the authors (LZZL) present a new semi-analytical solution for flow to a slanted well (including horizontal and vertical) of zero radius in a uniform anisotropic unconfined aquifer considering unsaturated flow above the water table. Details of the solution are included in supplementary material and appear to be correct. The underlying conceptual-mathematical model is similar to that of Tartakovsky and Neuman (2007) for a vertical well, with the exception that LZZL take the unsaturated zone to be finite (TN took it to be infinite). Their method of solution is somewhat different from that of TN. The authors evaluate their solution numerically and present it in the forms of time-drawdown curves in

*the saturated zone, synoptic profiles of dimensionless drawdown in both the saturated and the unsaturated zones, and flow rate across the water table, for a range of dimensionless parameters, concluding that the unsaturated zone has a significant effect on system behavior in all cases.*

*I find the paper to be clearly written and the mathematics well explained. I do, however, have a few fundamental questions to the authors:*

**Response:** Thank you for the positive comment.

*1. LZZL are aware that the conceptual-mathematical model of TN has been superseded by a more general model due to Mishra and Neuman (2010, 2011). In addition to having rendered the unsaturated zone finite (as do LZZL), the most important extensions introduced by MN are representations of unsaturated material properties by 4 (instead of 2) parameters, in a manner similar to that of Mathias and Butler (2006); accounting for storage in the pumping well (rather than treating this well as a line sink); and accounting for delayed response of piezometers and observation wells due to storage in these devices. TN have demonstrated that the four-parameter representation leads to more realistic estimates of aquifer parameters, based on observed drawdowns, than does the two-parameter model. They also demonstrated that storage of water in pumping and observation wells have significant impacts on drawdowns below and above the water table. My question to LZZL: Why have you not worked with a four-parameter model, and why have you not accounted for pumping and observation well storage?*

**Response:** The four-parameter model considering well storages may be more close to the realistic situation, and a model with more parameters naturally leads to better agreement between the model and observed data due to more flexible parameter fitting. This is the advantage of the four-parameter model considering well storages. However, a model with more parameters has its disadvantage as well. Firstly, it is more difficult to determine the values of those parameters precisely from a practical standpoint. Secondly, the predictive capability of a model with more parameters may not be better than that of a model with less parameters. For the discussion of this issue, one may consult the editorial messages of Voss (2011a, 2011b) and an interesting paper by Bredehoeft (2005).

This manuscript adopts a two-parameter approach because of the following reason.

In this study, we focus on the following questions: does the conclusions drawn for vertical wells also applicable for horizontal and slant wells when coupled unsaturated-saturated flow is of concern? Specifically, how important is the wellbore orientation on groundwater flow to a horizontal or slant well considering the coupled unsaturated-saturated flow process? To focus on answering these questions, we prefer to use a simpler model with the balance that keeping the most important physical process in the model but at the same time ignoring the secondly effects (such as wellbore effects of the pumping and observation wells). Indeed, the wellbore effects of the pumping and observation wells have introduced additional complexity to the solutions which are already substantially more complex than the solutions excluding the unsaturated zone process.

To avoid the influence of wellbore storage effects, we make the following proposal that could be implemented in the future investigations of coupled saturated-unsaturated flow process: using a pack system to insulate the screens of pumping and the observation wells, thus wellbore storage will not be a concern. Please see lines 154-165 and 251-257 of the revised text.

As far as we know, the two-parameter model can approximately represent unsaturated material properties to meet the need of this investigation. With all these being saying, we will certainly explore a four-parameter model in the future on the basis of this study, for the purpose of completion. However, such an effort deems to be a separate investigation that will be reported elsewhere.

*2. MN have demonstrated that the unsaturated zone may or may not have a significant impact on drawdown below the water table depending on the choice of system parameters and mode of observation. My question to LZZL: What justifies your blanket statement that the unsaturated zone has a significant effect on system behavior, without any qualifications?*

**Response:** Implemented. We revised our statements about the impacts of the unsaturated zone on drawdown in the revised manuscript.

1) The unsaturated flow has significant impact on drawdown curves in the saturated zone when $\kappa_D$ is less than 10 (the unsaturated-saturated system has a large retention capacity, a small initial saturated thickness, and/or a relatively small vertical hydraulic conductivity). The impact of unsaturated flow decreases as $\kappa_D$ increases, becoming small or insignificant when $\kappa_D$ close to $1 \times 10^3$. Please see lines 300-306 of the revised text.

2) The impact of unsaturated flow increases as dimensionless unsaturated thickness $b_D$ decreases. For the small $b_D (= 0.001)$, the drawdown curves approach the ZWP solution. For the large $b_D (= 100)$, the drawdown curves do not approach the ZZ solution because the impact of the unsaturated flow becomes significant at a fixed $\kappa_D$ of 0.1. Please see lines 327-333 of the revised text.

3) The effects of the unsaturated zone on the drawdown exist in any angle of inclination of a slant well, and this impact is more significant for the case of the horizontal well. The effects of the unsaturated zone on the drawdown are insensitive to the length of the horizontal well screen. The impact of the unsaturated zone decreases when the observation point moves downward, becoming further away from the unsaturated zone, as expected. Please see lines 351-355 of the revised text.

*3. MN have shown time-drawdown curves for the unsaturated zone; why have LZZL not done likewise? New developments in unsaturated zone sensor technology will likely make it practical, in the not too distant future, to observe unsaturated zone behavior at depth and use the MN (or LZZL) solution to interpret such observations quantitatively.*

**Response:** Implemented. We added Figs. 2c, 2d and 8b to illustrate time-drawdown behavior in the unsaturated zone. These graphs show that both the larger $\kappa_D$ and the larger

unsaturated thickness $b_D$ lead to the smaller drawdown in the unsaturated zone. We added these statements on lines 334-342.

*4. MN have used their solution to analyze published pumping test results, demonstrating (as already noted) that their new model provides more realistic parameter estimates than did earlier models, including that of TN (and hence, I conclude, LZZL). Why have LZZL not done the same?*

**Response:** Implemented. The application of our solution on a realistic pumping test is the best way to demonstrate the advantages of our solution. To the best of our knowledge, however, there are no field pumping tests specifically designed to test our model yet (slant or horizontal pumping well pumping test in an unconfined aquifer), which certainly can be an interesting research subject to explore in the future.

In order to demonstrate the parameter estimation advantage using our solution versus previous solutions (ZWP and ZZ solutions), we adopted Reviewer#1 and your suggestions and conducted a numerical simulation using the finite-element model COMSOL to generate synthetic drawdowns in saturated and unsaturated zones. After that, we used ZWP, ZZ and our solutions to fit these synthetic drawdowns and to assess the performance of these solutions. Please refer to our responses to the major comments 2 of Reviewer#1.

*5. MN have shown that their solution allows estimating unsaturated zone properties based on observed drawdowns in the saturated zone. Why have LZZL not attempted to do the same?*

**Response:** Implemented. Please refer to our responses to the major comment 2 of Reviewer#1 and your comment 4.

*In summary, the LZZL solution is interesting in that it is the first to consider slanted wells in the unconfined aquifer context. It however rests on a somewhat limited and outdated conceptual-mathematical model of the system; would it be possible for the authors to remedy this? In any case, the authors should show graphically how the unsaturated zone responds to pumping and, if at all possible, use their model to analyze real (or at the least synthetically generated) pumping test data.*

**Response:** Implemented. Please refer to our responses to your above comments.

**References**

Blumenthal, B. J., and Zhan, H. B.: Rapid computation of directional wellbore drawdown in a confined aquifer via Poisson resummation, Adv Water Resour, 94, 238-250, 10.1016/j.advwatres.2016.05.014, 2016.

Bredehoeft, J.: The conceptualization model problem--surprise, Hydrogeol J, 13, 37-46, 10.1007/s10040-004-0430-5, 2005.

Chen, C. S.: Analytical and Approximate Solutions to Radial Dispersion from an Injection Well to a Geological Unit with Simultaneous Diffusion into Adjacent Strata, Water Resour Res, 21, 1069-1076, Doi 10.1029/Wr021i008p01069, 1985.

Pechstein, A., Attinger, S., Krieg, R., and Copty, N. K.: Estimating transmissivity from single-well pumping tests in heterogeneous aquifers, Water Resour Res, 52, 495-510, 10.1002/2015wr017845, 2016.

Voss, C. I.: Editor's message: Groundwater modeling fantasies-part 2, down to earth, Hydrogeol J, 19, 1455-1458, 10.1007/s10040-011-0790-6, 2011a.

Voss, C. I.: Editor's message: Groundwater modeling fantasies -part 1, adrift in the details, Hydrogeol J, 19, 1281-1284, 10.1007/s10040-011-0789-z, 2011b.

Wang, Q., and Zhan, H.: Intra-wellbore kinematic and frictional losses in a horizontal well in a bounded confined aquifer, Water Resour Res, n/a-n/a, 10.1002/2015WR018252, 2016.

Wang, Q. R., and Zhan, H. B.: Radial reactive solute transport in an aquifer-aquitard system, Adv Water Resour, 61, 51-61, DOI 10.1016/j.advwatres.2013.08.013, 2013.

Zech, A., and Attinger, S.: Technical note: Analytical drawdown solution for steady-state pumping tests in two-dimensional isotropic heterogeneous aquifers, Hydrol Earth Syst Sc, 20, 1655-1667, 10.5194/hess-20-1655-2016, 2016.

Zhan, H. B., Wen, Z., and Gao, G. Y.: An analytical solution of two-dimensional reactive solute transport in an aquifer-aquitard system, Water Resour Res, 45, Artn W10501Doi 10.1029/2008wr007479, 2009a.

Zhan, H. B., Wen, Z., Huang, G. H., and Sun, D. M.: Analytical solution of two-dimensional solute transport in an aquifer-aquitard system, J Contam Hydrol, 107, 162-174, DOI 10.1016/j.jconhyd.2009.04.010, 2009b.